# Impact on Patient Management of a Novel Host Response Test for Distinguishing Bacterial and Viral Infections: Real World Evidence from the Urgent Care Setting

**DOI:** 10.3390/biomedicines11051498

**Published:** 2023-05-22

**Authors:** Boaz Kalmovich, Daniella Rahamim-Cohen, Shirley Shapiro Ben David

**Affiliations:** 1Health Division, Maccabi Healthcare Services, Tel Aviv 6812509, Israel; kalmovich@mac.org.il (B.K.); cohen_dani@mac.org.il (D.R.-C.); 2Faculty of Medicine, Tel Aviv University, Tel Aviv 6997801, Israel

**Keywords:** antibiotics, host response, diagnostics, uncertainty, bacterial, viral, infection

## Abstract

Antibiotic overuse and underuse are prevalent in urgent care settings, driven in part by diagnostic uncertainty. A host-based test for distinguishing bacterial and viral infections (MeMed BV) has been clinically validated previously. Here we examined how BV impacts antibiotic prescription in a real-world setting. The intention to treat with antibiotics before the receipt of a BV result was compared with practice after the receipt of a BV result at three urgent care centers. The analysis included 152 patients, 57.9% children and 50.7% female. In total, 131 (86.2%) had a bacterial or viral BV result. Physicians were uncertain about prescription for 38 (29.0%) patients and for 30 (78.9%) of these cases, subsequently acted in accordance with the BV result. Physicians intended to prescribe antibiotics to 39 (29.8%) patients, of whom 17 (43.6%) had bacterial BV results. Among the remaining 22 patients with viral BV results, antibiotic prescriptions were reduced by 40.9%. Overall, the physician prescribed in accordance with BV results in 81.7% of all cases (*p* < 0.05). In total, the physicians reported that BV supported or altered their decision making in 87.0% of cases (*p* < 0.05). BV impacts patient management in real-world settings, supporting appropriate antibiotic use.

## 1. Introduction

The appropriate use of antibiotics is fundamental to the proper management of patients with an acute infection. Diagnostic uncertainty regarding infection etiology is one of the major drivers of antibiotic misuse [1,2]. Identifying infectious etiology is a clinical challenge as both bacterial and viral etiologies may present with similar symptoms and clinical characteristics [3,4,5]. Recent studies have associated diagnostic uncertainty with a higher likelihood of antibiotics being overprescribed and especially broad spectrum antibiotics [6,7].

Antibiotic overuse (e.g., prescribing antibiotics to patients with viral infections) contributes to antimicrobial resistance (AMR) and is associated with unnecessary adverse events [3,6,8]. It is particularly burdensome in ambulatory settings [9]. A recent study estimated that the antibiotic overuse rate in US urgent care centers (UCC) is 46%, which is significantly higher than in Emergency Departments (EDs) (25%), and 2–3-fold higher than other outpatient settings [6]. On the other hand, antibiotic underuse (e.g., missing the diagnosis of a bacterial infection that may have benefited from antibiotic treatment) can lead to complications and negative patient outcomes [10].

A limited set of tests are available in the urgent care setting to help resolve diagnostic uncertainty. While routine tests for pathogen detection may aid in determining etiology, they suffer from limitations, such as long turnaround times, the detection of colonizing microorganisms and the inaccessibility of the infection site [11,12,13]. Another approach is the use of host biomarkers, such as C-Reactive Protein (CRP), procalcitonin, and white blood cell count. These have the advantage of short turnaround times and are not confounded by colonizers, yet their diagnostic performance is limited in distinguishing bacterial and viral infections [14,15,16,17,18].

MeMed BV^®^ (BV) is an FDA-cleared, rapid host response test designed to distinguish between bacterial and viral infections. It integrates the circulating levels of three immune proteins that change their expression differentially in response to an acute infection: the tumor necrosis factor-related apoptosis inducing ligand (TRAIL), the interferon-gamma induced protein-10 (IP-10), and CRP [19].

The ability of BV to accurately determine the infection etiology is attributed to the different yet complementary expression dynamics of its three constituent immune proteins. TRAIL, a member of the tumor necrosis factor family, plays a role in regulating programmed cell death, which is a critical response to viral infections. TRAIL’s expression is rapidly induced in viral infections and reduced in bacterial infections [19]. IP-10 is a small cytokine involved in multiple cellular processes, including chemotaxis and cell growth inhibition. IP-10’s expression is induced in response to bacterial infections and is more highly induced in response to viral infections [19]. Lastly, CRP is an inflammatory marker that is induced in response to multiple inflammation triggers, including bacterial infection. CRP typically takes 24–48 h to peak during infection development. The algorithm combining these three proteins, in effect, captures the immune response to infection mediated by multiple biological pathways, resulting in a robust and accurate read out of whether the host is responding to a bacterial versus viral infection.

BV’s diagnostic accuracy in differentiating bacterial and viral infections has been validated in multiple blind clinical studies, compared to an expert panel of adjudicators [11,14,19,20,21,22]. For example, Papan et al. demonstrated that BV could differentiate bacterial (including co-infection) infections from viral infections in children suspected to have a respiratory tract infection or those with fever without source, with a sensitivity of 94%, specificity of 94%, negative predictive value (NPV) of 99% and an equivocal rate of 10% [21]. Similar results were obtained by others, demonstrating a sensitivity of 92–94%, a specificity of 89–94% and NPV values of 93–99% [11,14,20,21,22]. Importantly, BV has demonstrated robustness across different age groups (adults and pediatric), days from symptom onset (0–7), comorbidities (e.g., hyperlipidemia and hypertension), clinical syndromes (e.g., upper respiratory tract infection, lower respiratory tract infection) and different pathogens (both viral, e.g., influenza types A and B and bacterial, e.g., group A streptococcus) [19].

Early evaluations have shown that the test can influence clinical decision making, but have focused on the ED setting and have not evaluated the pre-hospital, urgent care setting [23,24]. The application of BV in real-world settings, its adoption by physicians into the clinical workflow and its impact on patient management warrants further investigation. In this pilot study, we examined how BV impacts patient management, i.e., antibiotic prescription, at three urgent care centers. Physicians were also asked to report whether the test impacted their decision-making process.

## 2. Materials and Methods

### 2.1. Settings

The pilot study was conducted at three Maccabi Healthcare Services (MHS) Urgent Care Centers (UCCs). MHS is a non-profit Health Maintenance Organization in Israel, serving over 2.5 million citizens, and representing a quarter of the Israeli population. MHS operates ten outpatient UCCs nationwide. These centers provide urgent medical care after hours, i.e., between 19:00–23:00 on weekdays and 09:00–23:00 on weekends, for patients of all ages. The centers are equipped with on-site lab facilities and X-rays. The clinicians are registered, experienced physicians, and most of them (80%) are specialists in internal or family medicine or are pediatricians.

### 2.2. Study Design

This pilot study was designed to examine whether BV impacts patient management and antibiotic prescription at MHS UCCs. The study was designed to minimally interfere with the existing workflow at the UCC.

After an initial evaluation, including the patients’ history and a physical examination, each physician could refer eligible patients for a BV test at the UCC, along with additional tests such as blood and urine tests, a SARS-CoV-2 rapid antigen test, throat culture, and X-ray. Ordering the test was at the physician’s discretion, as part of the routine management of patients presenting with suspected bacterial or viral infections. Each physician was asked to fill out a questionnaire evaluating their intent to prescribe antibiotics before receiving the BV results (Pre-BV).

The BV results were made available alongside other lab findings and a clinical assessment. Antibiotic prescription practices were recorded in the medical file (Post-BV), and the physicians were asked to complete the second part of the questionnaire, evaluating the impact of BV on their decision-making process (Appendix A). Finally, the pre-BV and post-BV data were compared (Appendix A).

The study’s primary outcomes were assessed by comparing the pre-BV and post-BV data (Appendix A). The primary outcomes related to alignment between the BV result and antibiotic prescription practice are as follows:Alignment between the BV result and antibiotic prescription in cases where the physician reported diagnostic uncertainty.Alignment between BV viral result and antibiotic prescription in cases where the physician reported intent to prescribe (potentially unwarranted antibiotic prescriptions).Alignment between the BV bacterial result and antibiotic prescription in cases where the physician reported no intent to prescribe (potentially missed bacterial infections).

The secondary outcome related to the physician’s report on BV’s impact: The proportion of cases in which the physician reported that BV impacted their decision-making process regarding patient management

### 2.3. Data Collection

The data collected included demographics, symptoms and physical examination, comorbidities, laboratory parameters, and antibiotic prescription. Furthermore, the clinical assessment and follow up data were performed and collected for each case by one of the authors (SBBD). Medical records were retrieved 7 days following the UCC visit and reviewed for subsequent hospitalization, emergency room visits, primary care physician visits and antibiotic use.

The study was approved by the institutional review board (IRB) of MHS, number MHS-0138–21. Informed consent was waived by the IRB; identifying details of the participants were removed before computational analysis.

### 2.4. MeMed BV Test Measurements

BV (MeMed BV^®^, MeMed, Haifa, Israel) can be used for adult and pediatric patients presenting to EDs and UCCs with a suspected acute bacterial or viral infection, who have had symptoms for less than seven days. The test generates a numeric score that falls within discrete interpretation bins based on the increasing likelihood of bacterial infection. BV score < 35 indicates viral (or other non-bacterial) infection; 35 ≤ BV score ≤ 65 is an equivocal result and BV score > 65 indicates a bacterial infection or bacterial-viral co-infection (Figure 1). Equivocal results represent valid test results wherein BV does not add further information in addition to the physician’s infection etiology diagnosis.

Physicians were guided to order the test in accordance with the regional indications for use. The intended use of BV is in conjunction with clinical assessments and other laboratory findings as an aid to differentiate bacterial from viral infection.

BV tests were performed at the UCC according to the manufacturer’s instructions using a rapid immunoassay platform (MeMed Key^®^, MeMed, Haifa, Israel). Test results were available within 15 min from serum samples and were immediately provided to the treating physician. BV results were documented in the patient’s electronic medical record.

### 2.5. Statistical Analysis

Based on the well-documented rate of antibiotic overuse in the urgent care settings, and an expected high rate of viral diseases, it was hypothesized that >70% alignment between BV results and clinical practice would constitute a significant proportion. Similarly, it was hypothesized that if 1-in-5 (>20%) physicians with an intention to prescribe antibiotics changed their decision after receiving viral BV results, this would constitute a significant reduction. Lastly, it was hypothesized that if >70% of physicians indicated that the BV test impacted their decision-making process (either supported or changed), this would constitute a significant contribution to patient management.

A one-sample, one-sided proportion test was used to evaluate the statistical significance of the alignment between the test results and clinical practice, and the impact of the test according to physicians. The 90% two-sided confidence interval (CI90%) was calculated using the adjusted Wald method [25]. The 95% lower confidence bound was then used to test if the observed proportion was significantly higher than 70% or 20% (as per hypothesis). Python package statsmodels version 0.12.2 (https://www.statsmodels.org/stable/index.html) was used for statistical analyses.

## 3. Results

### 3.1. Study Cohort

Between December 2020 and May 2021, 475 BV tests were ordered at the physician’s discretion and for 161 of the patients, the physician filled in a questionnaire. Nine of these 161 enrolled patients were excluded either because of incomplete questionnaires or due to non-infectious etiology (Figure 2).

The resulting study cohort of 152 patients included 88 children (median age 1.7 years (interquartile range (IQR) 1.0–2.9) and 64 adults (median age 43 years (IQR 30.7–55.8)). Among adults, there were 12 elderly patients (aged 65 or older), with a median age of 75.6 (IQR 73.3–80.3). Antibiotics were prescribed at the UCC to 62 (40.8%) patients. The most common discharge diagnosis was fever (*n* = 51, 33.6%), followed by viral infection (*n* = 25, 16.4%), pneumonia and upper respiratory infection (*n* = 13, 8.6% for both). Seven-day follow up data indicated that 11.2% (17/152) of patients presented to the ED within a week of discharge and 7.2% (11/152) were hospitalized (Table 1).

Microbiological tests were ordered for 92/152 patients, including 16 SARS-CoV-2 tests (all negative), 15 throat cultures (3 returned positive), 29 blood cultures (all negative), and 38 urine cultures (6 returned positive). Serological tests for Epstein–Barr virus (EBV) were ordered for 3 patients (2 returned positive), and Cytomegalovirus (CMV) was ordered and detected in one patient. In total, 13/161 patients had a pathogen detected.

The characteristics of the study population broken down based on intention to prescribe antibiotics (uncertain, intention to prescribe and no intention to prescribe) are provided in the Appendix A.

### 3.2. Impact of BV Result in Cases of Diagnostic Uncertainty

The intention to treat with antibiotics before the receipt of a BV result was compared with practice after the receipt of a BV result in order to assess the impact of BV on patient management (Figure 2). Since an equivocal result is non-informative for etiology and not intended to influence patient management, data relating to the 21 patients with equivocal results are presented separately (Appendix A). Physicians were uncertain whether to prescribe antibiotics for 29.0% of the 131 patients (38/131) with a bacterial or viral BV result. Among these, seven patients received bacterial BV results, and all were subsequently prescribed antibiotics; six at the UCC and one was referred to the ED for further evaluation and antibiotic treatment. The remaining 31/38 patients received a viral BV result and physicians did not prescribe antibiotics for 74.2% (23/31) of cases. Among these 23 patients, only 3 were prescribed antibiotics within 7 days and 1 was hospitalized (Appendix A). Notably, two of these three are outside BV’s indication for use (one presented without fever and one with symptoms for 7 days). Overall, the physicians prescribed in accordance with the BV result in 78.9% (30/38) of cases with diagnostic uncertainty.

### 3.3. Impact of BV Result in Cases with Intention to Prescribe

Physicians intended to prescribe antibiotics to 29.8% (39/131) of patients (Figure 2). Among these, 56.4% (22/39) received a viral BV result. Physicians acted in accordance with the BV test in 40.9% (9/22) of cases and did not prescribe antibiotics, changing their patient management (CI90%: 25.6–58.2%, *p* < 0.05). None of these patients developed complications within 7 days of discharge and only one was prescribed antibiotics within 7 days (Appendix A).

### 3.4. Impact of BV Result in Cases with no Intention to Prescribe

Physicians did not intend to prescribe antibiotics to 41.2% (54/131) of patients (Figure 2). Of these, 14.8% (8/54) received a bacterial BV result and subsequently all were treated with antibiotics, representing potentially missed bacterial infections. Notably, physicians reported that the BV result affected their decision-making process for all eight cases. Two of these patients were hospitalized and discharged with a diagnosis of pneumonia, one of whom was elderly (Appendix A). Another of these eight patients were elderly and discharged with a diagnosis of pyelonephritis. All elderly cases are presented in Appendix A. Three (6.5%) patients received antibiotics despite no intention to prescribe and viral BV results.

Overall, the physicians prescribed in accordance with the BV result in 81.7% (107/131) of all cases in the study (CI90%: 75.5–86.6%, *p* < 0.05).

### 3.5. Impact of BV Result on Decision Making Process According to the Physician

In addition to the BV results, physicians had access to X-ray results, blood tests and other supporting information prior to making their final prescription decision. To better understand the impact specifically of BV on their decision-making process, physicians were asked to report whether the test impacted their patient management decision. Physicians reported that BV changed their decision regarding antibiotic treatment for 21.4% (28/131) of patients. Furthermore, physicians reported that BV supported their decision making in an additional 65.6% (86/131) of cases and did not affect them in 13.0% (17/131) of the cases (Figure 3). Overall, physicians reported BV had a positive impact on patient management in 87.0% of cases in which BV results were either bacterial or viral (114/131, CI90%: 81.4–91.2%, *p* < 0.05) and for 100.0% of the twelve elderly patients.

## 4. Discussion

This pilot study evaluated the impact on patient management of a new host response test, BV, for discriminating bacterial and viral infections deployed at three urgent care centers. Both pediatric and adult patients were enrolled, with BV ordered at the physician’s discretion. The impact was evaluated by comparing the physician’s intention to treat with antibiotics prior to receiving the BV result, and the antibiotic prescription as documented in the medical record after receiving the BV result. Physicians not only prescribed in accordance with the BV result for most cases when they were diagnostically uncertain, but also changed their prescription decision for all cases with bacterial BV results and for 40.9% of viral BV results. When asked about the utility of BV, they reported that it supported or changed their antibiotic prescription decisions in 87.0% of cases and did not impact them in 13.0%. The antibiotic prescription patterns combined with the physician’s reported impact provide evidence that BV aids in patient management in the UCC.

A feature of the BV test is that it typically provides bacterial or viral results for 87–92% of patients, with equivocal results only in 8–13% of cases. In the present study, 21/152 of patients received equivocal results (13.8%), falling within the expected range. In cases in which the BV result is equivocal, physicians are guided to employ other complementary patient data in their decision making for patient care. Accordingly, an equivocal result would not be anticipated to cause the physician to misdiagnose or mismanage patients.

Three patients were prescribed antibiotics despite the physician not intending to prescribe (Pre-BV) and having viral BV results. None of these patients had any microbiologically confirmed infection (negative cultures) and it is difficult to comment on whether the antibiotics were warranted. These cases highlight a well-known challenge of diagnosing infection etiology, namely that in many cases, a clinically relevant pathogen cannot be detected or directly linked to the disease [5,26]. Therefore, given the frequent absence of a gold standard for infection etiology, we took the approach of examining patient outcomes as an indicator by which to evaluate the real-world impact of BV.

Here, we show that in the patients for whom antibiotics were initially intended but were eventually not prescribed, none manifested sequelae in a 7-day follow up. Moreover, integrating BV into the clinical workflow identified 8 potentially missed bacterial infections. These eight cases included elderly patients and patients eventually diagnosed with community acquired pneumonia, for whom the prompt initiation of antibiotic therapy is especially important [10,27]. Taken together, these findings are the first demonstration that BV can be used to safely guide antibiotic prescription, and merit validation in larger studies.

The main strength of this pilot study is its focus on the real-world use of BV. With this in mind, the study was designed to minimally interfere with the day-to-day work of the UCC. Accordingly, the test was integrated into the clinical workflow and tests were ordered at the physicians’ discretion. Another strength is that several UCCs participated in the study, with multiple physicians at each site using BV. As both children and adults were enrolled, physicians of different specialties participated. A limitation is that only patients managed by physicians who chose to complete the pre- and post-test questionnaire were enrolled in the study. While this design ensured minimal interference with routine care, it may have selected early adopter physicians and introduced biases, such as availability bias. Indeed, although every effort was made for the questionnaires not to incur a burden and interrupt clinical workflow, in the busy setting of the UCC, the request to fill out a form is a distraction. This can be understood from the observation that there were completed questionnaires for only 156 out of the 475 patients for whom BV was ordered. Future studies should attempt to address this limitation by minimizing and automating the questionnaires.

As a pilot study, the sample size was relatively small. Future studies with larger cohorts of patients, with the possibility of performing separate sub analyses for children and adults, are warranted.

## 5. Conclusions

BV contributed to physician’s decision-making process in the management of patients with acute infections. These pilot results demonstrate the applicability of BV in the UCC workflow and support its clinical utility regarding appropriate antibiotic use. Additional real-world evaluations are warranted.

## Figures and Tables

**Figure 1 biomedicines-11-01498-f001:**
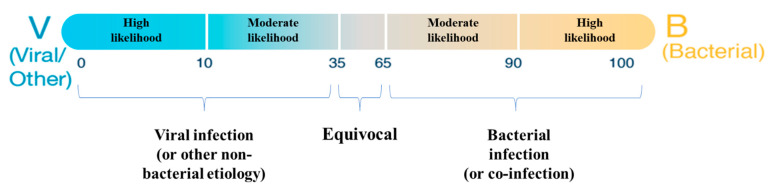
BV test results and interpretation bins according to the manufacturer’s instructions for use.

**Figure 2 biomedicines-11-01498-f002:**
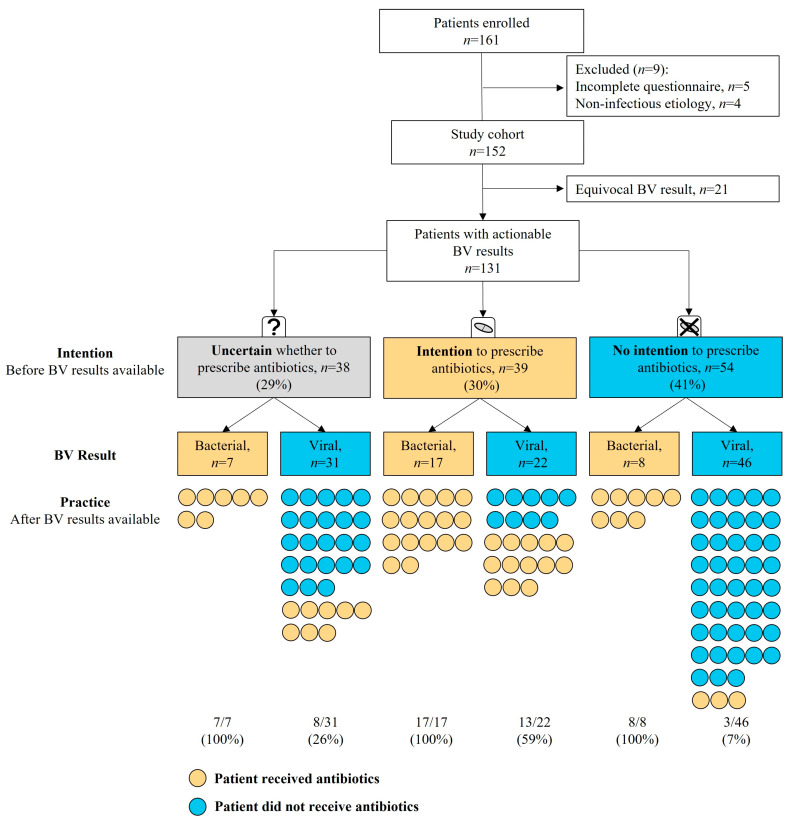
Patient enrolment flow, the physician’s intention to prescribe antibiotics after initial examination (Intention), the BV result and whether antibiotics were prescribed according to the medical records (Practice). Practice is reported as the number of patients for whom antibiotics were prescribed (orange) or not prescribed (blue); the prescription rate is in brackets at the bottom.

**Figure 3 biomedicines-11-01498-f003:**
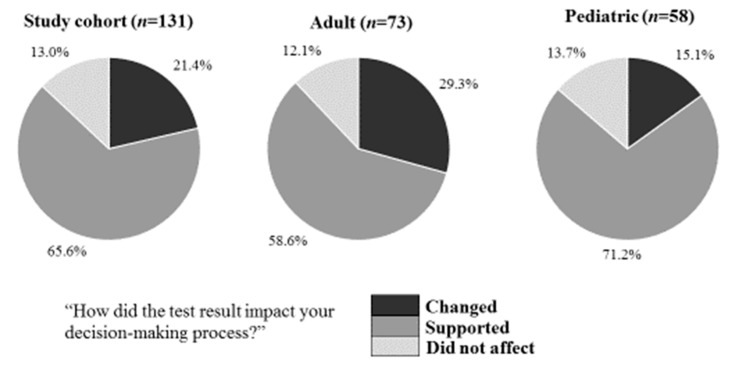
Distribution of answers to the post-test question “How did the test result impact your decision-making process?” for the cases with a bacterial or viral BV result.

**Table 1 biomedicines-11-01498-t001:** Characteristics of the study cohort.

Age (Median, IQR)	All (*n* = 152)	6.0 (1.5–38.3)
	Children (*n* = 88) Adults (*n* = 64) Sub-cohort: Elderly; ≥65 (*n* = 12)	1.7 (1.0–2.9) 43.0 (30.7–55.8) 75.6 (73.3–80.3)
Sex	Female	77 (51%)
Acute illness	Days from symptoms onset (Median, IQR)	2 (1–4)
	Prescribed antibiotics at UCC	59 (39%)
Discharge Diagnosis	Fever	51 (34%)
	Viral Infection	25 (16%)
	Pneumonia Upper Respiratory Infection Urinary Tract Infection/Pyelonephritis Tonsillitis/Pharyngitis Other *	13 (9%) 13 (9%) 12 (8%) 10 (7%) 28 (18%)
Follow up (7 days)	Hospitalized	11 (7%)

* Other diagnoses included three or fewer cases of: Cough, sore throat, flank pain, acute bronchitis, weakness, stomatitis, scarlet fever, dizziness, headache, dysuria, low back pain, gastroenteritis, parotitis, fatigue, hip pain with fever, laryngotracheitis, diarrhea, renal colic, observation, and occult bacteremia (suspected). IQR: interquartile range, UCC: urgent care center.

## Data Availability

Not applicable.

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
