# Peer review of "Impact on Patient Management of a Novel Host Response Test for Distinguishing Bacterial and Viral Infections: Real World Evidence from the Urgent Care Setting"

_biomedicines, 2023, doi:10.3390/biomedicines11051498_

Round 1

Reviewer 1 Report

The paper is an important study that should improve the quality of medicine.  My only suggestion is that the paper should include more information on why the three different immune proteins were chosen.  While references are provided, the theory behind the choice would be a useful addition to the paper and give a better understanding to the assay.

Reviewer 2 Report

 This study presents the impact of a medical test that aims to help clinicians in deciding whether somministrate or not antibiotics to patients with a suspected infection. Is an interesting study in that would avoid the overuse and missuse of antibiotics when the clinicians are not sure whether the patient has a bacterial or viral infection. The test needs serum from the patient and takes 15 min to be performed. 

My main concern regards the description of the test which is missed in the paper; authors should explain more in detail what is the basis of the test and how this is functioning; it is also not clear why 21 cases have an undetermined result and how this can be solved. 

It would be also interesting to know why some clinicians did not accept to use the test which is not interfering with the normal flochart of the patient assistance
